# An examination of psychometric properties of study quality assessment scales in meta-analysis: Rasch measurement model applied to the firefighter cancer literature

**Soyeon Ahn**[1]*, **Paulo S. Pinheiro**[2], **Laura A. McClure**[2,3], **Diana R. Hernandez**[3], **Alberto J. Caban-Martinez**[2,3], **David J. Lee**[2,3]

1 Department of Educational and Psychological Studies, School of Education and Human Development, University of Miami, Miami, Florida, United States of America, 2 Sylvester Comprehensive Cancer Center, Leonard M. Miller School of Medicine, University of Miami, Miami, Florida, United States of America, 3 Department Public Health Sciences, Leonard M. Miller School of Medicine, University of Miami, Miami, Florida, United States of America

* s.ahn@miami.edu

**Data Availability Statement:** All relevant data are within the paper and its Supporting information files.

## Abstract

Most existing quality scales have been developed with minimal attention to accepted standards of psychometric properties. Even for those that have been used widely in medical research, limited evidence exists supporting their psychometric properties. The focus of our current study is to address this gap by evaluating the psychometrics properties of two existing quality scales that are frequently used in cancer observational research: (1) Item Bank on Risk of Bias and Precision of Observational Studies developed by the Research Triangle Institute (RTI) International and (2) Newcastle-Ottawa Quality Assessment Scale (NOQAS). We used the Rasch measurement model to evaluate the psychometric properties of two quality scales based on the ratings of 49 studies that examine firefighters' cancer incidence and mortality. Our study found that RTI and NOQAS have an acceptable item reliability. Two raters were consistent in their assessment, demonstrating high interrater reliability. We also found that NOQAS has more items that show better fit than the RTI scale. The NOQAS produced lower study quality scores with a smaller variation, suggesting that NOQAS items are much easier to rate. Our findings accord with a previous study, which conclude that the RTI scale was harder to apply and thus produces more heterogenous quality scores than NOQAS. Although both RTI and NOQAS showed high item reliability, NOQAS items are better fit to the underlying construct, showing higher validity of internal structure and stronger psychometric properties. The current study adds to our understanding of the psychometric properties of NOQAS and RTI scales for future meta-analyses of observational studies, particularly in the firefighter cancer literature.

## Introduction

Assessment of study quality is a critical aspect of conducting meta-analyses. Study quality considerably varies across studies and may lead to heterogeneity in study findings [1–5]. Bérard

**Funding:** This study was supported by funds from Florida State Appropriation #2382A (Principal Investigator: Kobetz). Research reported in this publication was also supported by the National Cancer Institute of the National Institutes of Health under Award Number P30CA240139. The funders had no role in study design, data collection and analysis, decision to publish, or preparation of the manuscript.

**Competing interests:** The authors have declared that no competing interests exist.

and Bravo warned that overall effect size estimates obtained from meta-analysis that do not account for variation in study quality may suffer from increased Type I error rates [6]. In addition, other factors investigators ought to be concerned when evaluating studies include the sources, directions, and even plausible magnitudes of such biases [7, 8]. Therefore, many researchers suggest that the quality of primary studies should be accurately assessed and used in meta-analysis [5, 7]. Despite the importance of assessing study quality in general, many researchers have identified challenges in dealing with the quality of primary studies in meta-analyses [3, 6, 9]. One of the critical issues is that while there are a variety of scales to assess the quality of primary studies, none has been universally adopted [10]. In fact, there is no consensus about how study quality should be conceptualized or measured in the existing quality scales [5, 11–13]. Moreover, most existing quality scales have been developed with minimal attention to accepted standards of psychometrics properties such as reliability and validity [14]. Most of the research has focused on interrater reliability measures, such as kappa statistics or percentage of agreement, rather than item reliability, content validity, or construct validity. In addition, even for those that have been used widely in medical research, no to little evidence exists supporting their psychometric properties.

Therefore, the focus of our current study is to address this gap by evaluating the psychometrics properties (i.e., item reliability, interrater reliability, and construct reliability) of existing quality scales that are frequently used in cancer observational research: (1) Item Bank on Risk of Bias and Precision of Observational Studies developed by the Research Triangle Institute (RTI) International [15] and (2) Newcastle-Ottawa Quality Assessment Scale (NOQAS) [16]. Specifically, we used the Rasch measurement model [17] to evaluate the psychometric properties of these two quality scales based on the ratings of 49 studies that examine firefighters' cancer incidence and mortality. The present study is focused on three primary research questions, namely:

1. Can the RTI or the NOQAS scale be considered reliable?

2. Do the items of RTI or NOQAS fit the overall quality score?

3. Do the individual studies fit the overall quality score?

## Study quality

Two different frameworks have been proposed in the literature to define and measure the quality of primary studies in meta-analysis [18]. One is based on the validity framework developed by Campbell and his associates [19] and the other, called "quality assessment", was proposed by Chalmers and his colleagues [8].

The former approach, based on the idea of Campbell and his associates, suggests a matrix of designs and their features or threats to validity. The validity framework includes 33 separate threats to validity based on four distinct categories: internal, external, statistical, and construct validity [18]. This validity framework for assessing the quality of primary studies in a meta-analysis is mainly used in the social sciences. For instance, Devine and Cook [20] evaluated the quality of primary studies based on the validity framework by examining six design features representing internal, external, statistical, and construct validity (e.g., floor effect, publication bias, attrition, and domains of content).

The second approach, proposed by Chalmers and his associates, has been applied primarily to medical research [8, 18, 21–24]. The objective of Chalmers' system is to quantify the overall quality of primary studies based on in-depth criteria for assessing randomized controlled trials. Chalmers and his colleagues mainly focused on construct validity and statistical conclusion

validity, examining such features as randomization, blinding of the statistician, and minimization of data-extraction bias [18].

## Study quality assessments in observational studies

An informal PubMed search of published meta-analyses and systematic reviews in the cancer literature revealed that the Newcastle-Ottawa Quality Assessment Scale (NOQAS) [16] was the widely employed tool for review articles which focused on risk factor association studies [25–31]. This tool was employed in a recent meta-analysis of the firefighter cancer literature [32]. The second identified assessment tool was less commonly employed in cancer-focused meta-analyses and systematic reviews [33, 34]: the Research Triangle Institute (RTI) International and Item Bank on Risk of Bias and Precision of Observational Studies [35, 36]. Although not commonly employed in cancer meta-analyses [33], it has been utilized in a variety of syntheses of other disease outcome association studies [36–41] and was employed in a systematic review of lung function in firefighters [42]. Of note, some investigators have employed both the NOQAS and the RTI item bank to assess quality in meta-analyses and systematic reviews [37, 42, 43]. The RTI item bank is comprised of 29 multiple-choice questions that is designed to assess a range of risk of bias and precision domains for a variety of observational study designs [36, 37]. These domains include: sample definition and selection, interventions/exposure, outcomes, creation of treatment groups, blinding, soundness of information, follow-up, analysis comparability, analysis outcome, interpretation, and presentation and reporting. Investigators are encouraged to select items from the bank that are most appropriate to the content area and study design of studies under assessment.

The 8-item NOQAS was developed to assess the quality of nonrandomized studies with specific assessment forms for case-control and cohort study designs [16]. Several questions are designed to be tailored for use given the content being assessed. A simple summary quality score can be obtained by summing each individual item judged to be of high quality, although given its relatively short length investigators often report quality levels for the individual 8 items for each study under review. The NOQAS has been recommended for use for the assessment of quality of observational study designs [41, 44].

Our literature search using PubMed, PsycInfo and Medline resulted in one published study that compares the psychometric evidence of NOQAS to RTI. In a study by Margulis and her colleagues [40], two raters independently assessed the quality of 44 primary studies with RTI and NOQAS. After coding the quality of studies, Margulis and her colleagues computed interrater agreement using percentage of agreement and the first-order agreement coefficient statistics. In their study, the relationship between NOQAS and RTI for ranking ordering studies in terms of risk of bias was found to be medium, as indicated by the Spearman's rank correlation coefficients of .35 and .38. Also, authors stated that NOQAS is easier to apply than the RTI item bank, but more limited in its scope, although the scope of quality was similar between NOQAS and RTI. Lastly, the interrater reliabilities between raters were reported to be fair for both NOQAS and RTI.

Like a study by Margulis and her colleagues [40], a few published studies addressed the quality of either NOQAS or RTI, using interrater reliability measures such as kappa statistics or percentage of agreement between raters [41, 44]. Likewise, all these studies evaluated the psychometric properties of the quality assessment tools under the Classical Test Theory (CTT) framework, which is somewhat simple by analyzing the raw data of the instrument. Also, most of the existing studies focused on interrater reliability or face validity of items used to measure the quality of individual studies.

## Rasch measurement model

Whereas classical test theory (CTT) has been frequently used in evaluating the validity and reliability of study quality ratings, some issues have arisen regarding the calibration of item difficulty, sample dependence of coefficient measures, and estimates of measurement error. The Rasch model enables us to address those issues by (1) assessing the dimensionality of assessment; (2) identifying redundant items or items that measure a different construct or construct-irrelevant factors through the item-fit; (3) identifying items that should be flagged based on their difficulty levels; and (4) assessing whether response categories are appropriate for distinguishing items by their quality.

The Rasch Measurement Theory (RMT) is a psychometric model to analyze categorical data (particularly dummy variables) as a function of the person (e.g., rater or reviewer)'s ability on a trait and the item difficulty [17]. Andrich [45] then developed the Rasch Rating Scale Model (RSM, also called the Polytomous Rasch model) for polytomous data, which is data with more than two ordinal categories. The RSM provides estimates of person location on a continuous latent variable ($a$), item difficulties ($b$), and an overall set of thresholds that are fixed across items ($c$).

RMT obtains information from the person and the item to estimate the probability of a person with a given level of ability to answer a given item correctly, thus, connecting person ability to item difficulty [46]. This probabilistic framework allows RMT to be falsifiable and to meet the linearity assumptions of parametric statistical tests. Therefore, measures of fit statistics for both person-fit and item-fit can be obtained, which provide evidence of validity—how well the model can predict the response to each item.

In addition, RMT transforms ordinal data to logits, which allows a proper use of parametric statistical analysis, without assumption violation that is associated with Type I and II error inflation. Lastly, the item parameters estimated by RMT are generally invariant to the population used to generate these estimates. In other words, parameter estimates obtained from a sufficient sample should be equivalent to those obtained from another sufficient sample despite the average person's ability level in each of the samples [46]. This property of RMT allows for greater generalization of results as well as more sophisticated applications.

## Psychometric properties in Rasch measurement model

For any quality test or assessment, the supporting evidence must have three psychometric properties—validity, reliability, and fairness [47]. This section briefly reviews how each of these psychometric properties can be assessed when using the RMT. In this study, our focus is on reliability and validity.

**Reliability.**   Reliability refers to the consistency or precision of scores across replications of a testing procedure. Under RMT, the Rasch-based reliability index, called the *reliability of separation*, is used to measure the reliability of a test or assessment. A reliability of separation index is obtained based on latent measures with equal intervals along the underlying continuum and it reflects how distinct latent scores are along the scale, which ranges from 0 to 1. This is defined as:

$$Reliability = \frac{SD^2 - MSE}{SD^2} \tag{1}$$

, where $SD$ = standard deviation of Rasch measures of a specific facet (e.g., students, tasks, and raters) and $MSE$ = average Mean Squared Errors of Rasch measures for each facet. Higher values indicate higher reliability. High reliabilities are preferred because they indicate a good presentation of Rasch measures across the entire range of the latent scale.

**Table 1. Fit categories for interpreting infit and outfit mean square errors.**

| Mean Square Residual (MnSq) | Interpretation | Fit Category |
|---|---|---|
| $0.5 \leq MnSq \leq 1.5$ | Productive for measurement | A |
| $MnSq < 0.5$ | Less productive for measurement, but not distorting of measures | B |
| $1.5 < MnSq \leq 2.0$ | Unproductive for measurement, but not distorting of measures | C |
| $2.0 < MnSq$ | Unproductive for measurement, distorting of measures | D |

**Validity.** Validity refers to the degree to which theory supports the interpretation of test scores[47]. Under the RMT, the Infit and Outfit Mean Square (MnSq) statistics can be used to evaluate how well the measures of an individual facet (i.e., item, study, and rater) fit the constructed latent scale (i.e., study quality score). In particular, the Infit MnSq identifies irregular response patterns, and the Outfit MnSq detects large residual values. The expected value for both Infit and Outfit MnSq statistics is 1.0, which shows a perfect fit to the underlying scale. The fit indices provide diagnostic information for identifying misfit elements on each facet (e.g., item, study, or rater), supporting the validity arguments of internal structure. Therefore, the validity can be rated on a scale ranging from A (item, study, or rater fits the scale very well) to D (item, study, or rater does not fit the scale). See Table 1 for the guidelines for interpreting the Infit and Outfit MnSq values.

## Methods

### Description of 49 studies on firefighter cancer incidence and mortality

The studies evaluated in this quality assessment were gathered for a meta-analysis project that examines cancer incidence and mortality risk among firefighters. The included studies were identified through a comprehensive literature search using multiple databases including ERIC, PsycINFO, ProQuest Dissertation & Theses, PUBMED, and MEDLINE via EBSCO, and online search engines including Embase, Web of Science Core collection, Google Scholar, and SCOPUS. A total of 49 studies were identified that met the inclusion and exclusion criteria.

Two independent raters were responsible for coding (1) study design characteristics, (2) outcome type, (3) cancer coding system, (4) cancer types, (5) source of occupation designations, (6) type of incident that firefighters attended, (7) sample characteristics, and (8) study characteristics. Two additional reviewers were responsible for coding the statistical estimates presented in these studies for computing a standardized incidence ratio and (2) a standardized mortality ratio.

### Procedure

Two content experts on epidemiology independently rated 49 observational studies using RTI item banks and NOQAS. Two independent raters are: (1) a cancer epidemiologist who holds a PhD in Epidemiology and has 20 years of experience in cancer research and teaching; and (2) a chronic disease and occupational epidemiologist who holds PhD in preventive medicine and community health and has over 30 years of teaching and research experience and the Principal Investigator of the Florida cancer registry (Florida Cancer Data System).

The two study quality scales were first tested by the independent reviewers on a sample of studies (i.e., random sample of 5–7 studies) to ensure that consistent assumptions and criteria were employed by raters. Slight modifications were then made to the original quality assessments to better align with the methods of the studies evaluated, and some items were removed that were not relevant. The items evaluated along with their modifications (modifications are italicized) and specific instructions (13 RTI and 8 NOQAS items) are displayed in Table 2.

**Table 2. Revised risk of bias and precision of observational studies & Newcastle-Ottawa quality assessment scales.**

| SECTION | STATEMENT, INSTRUCTIONS & SCORING |
|---|---|
| **NEWCASTLE-OTTAWA QUALITY ASSESSMENT SCALE** | |
| Selection | 1. **Representativeness of the exposed cohort—Newcastle1**<br>Item is assessing the representativeness of exposed individuals in the community, not the representativeness of the sample of women from some general population.<br>a) truly representative of the average _______________ (describe) in the community = 1 star<br>b) somewhat representative of the average _____________ in the community = 1 star<br>c) selected group of users eg nurses, volunteers = 0 stars<br>d) no description of the derivation of the cohort = 0 stars |
| | 2. **Selection of the non-exposed cohort—Newcastle2**<br>a) drawn from the same community as the exposed cohort = 1 star<br>b) drawn from a different source = 0 stars<br>c) no description of the derivation of the non-exposed cohort = 0 stars. *Note: In the case of general population can code = 1 star; if other occupational groups only then code 0.5 star given possible overlapping exposures.* |
| | 3. **Ascertainment of exposure—Newcastle3**<br>a) secure record (e.g. surgical records) = 1 star<br>b) structured interview = 1 star<br>c) written self report = 0 stars<br>d) No description = 0 stars<br>*Note: if self-report = 0 stars. Exposure based on registry/death records = 0.5 star. Disregard other confounders when assessing on this item.* |
| | 4. **Demonstration That Outcome of Interest Was Not Present at Start of Study—Newcastle4**<br>*In the case of mortality studies, outcome of interest is still the presence of a disease/ incident, rather than death. That is to say that a statement of no history of disease or incident earns a star.*<br>a) yes = 1 star<br>b) no = 0 stars<br>*Note: For mortality studies code = 1 star if there if you can assume that persons were alive at enrollment into the cohort.* |
| **COMPARABILITY** | 1. **Comparability of Cohorts on the Basis of the Design or Analysis—Newcastle5**<br>Age and one other control OR stratified analysis will qualify for two star rating<br>a) study controls for _______ (select the most important factor) = 1 star<br>b) study controls for any additional factor = additional 1 star |
| **OUTCOME** | 1. **Assessment of outcome—Newcastle6**<br>a) independent blind assessment (Independent or blind assessment stated in the paper, or confirmation of the outcome by reference to secure records = 1 star<br>b) record linkage (e.g. identified through ICD codes on database records) = 1 star<br>c) self-report (i.e. no reference to original medical records to confirm the outcome) = 0 stars<br>d) no description = 0 stars<br>*Note: ICD version should be specified in order to earn 1 star.* |
| | 2. **Was follow-up long enough for outcomes to occur—Newcastle7**<br>An acceptable length of time should be decided before quality assessment begins<br>a) yes (select an adequate follow up period for outcome of interest) = 1 star<br>b) no = 0 stars<br>*Note: Must mention > = 2 year lag; if not then assign = 0 stars* |
| | 3. **Adequacy of follow up of cohorts—Newcastle8**<br>This item assesses the follow-up of the exposed and non-exposed cohorts to ensure that losses are not related to either the exposure or the outcome.<br>a) complete follow up—all subjects accounted for = 1 star<br>b) subjects lost to follow up unlikely to introduce bias = 1 star<br>c) low follow up rate or no description of those lost = 0 stars<br>d) no statement = 0 stars<br>*Note: No star assignment if loss to follow-up exceeds 10%; Active follow-up (e.g., last date of contact reported when the event could be verified) = 1 star; passive follow-up or no mention of follow-up = 0 stars)* |
| **RISK OF BIAS AND PRECISION OF OBSERVATIONAL STUDIES (RTI)** | |

(*Continued*)

**Table 2.** (Continued)

| SECTION | STATEMENT, INSTRUCTIONS & SCORING |
|---|---|
| **SAMPLE DEFINITION AND SELECTION** | **Is the study design prospective, retrospective, or mixed?** [*Abstractor*: *Prospective design requires that the outcome has not occurred at the time the study is initiated and information is collected over time to assess relationships with the outcome (and includes nested case-control studies). Mixed design includes case-control or cohort studies in which one group is studied prospectively and the other retrospectively. A retrospective design analyzes data from past records. The question is not applicable to cross-sectional studies.* <br> *Note: retrospective cohort designs should be coded as retrospective*] <br> Prospective / Mixed / Retrospective / Cannot determine or not applicable |
| | **Are critical inclusion/exclusion criteria clearly stated (does not require the reader to infer)?—RTI1** [*Principal Investigator (PI): Provide direction to abstractors by listing individual criteria of a priori significance and minimal requirements for criteria to be considered "clearly stated." Include this question to identify specific inclusion/exclusion criteria that should be consistently recorded across studies*] [*Abstractor*: *Use "Partially" if only some criteria are stated or if some criteria are not clearly stated (corresponding to directions provided by the PI). Note that studies may describe inclusion criteria alone (i.e., include x), exclusion criteria (i.e., do not include x), or a combination of inclusion and exclusion criteria.* <br> *Note: many studies will describe these criteria on the basis of identifying firefighters thru employment/ certification/death certificate coding records which can be classified as 'yes'*] <br> Yes / Partially: some, but not all, criteria stated or some criteria not clearly stated / No |
| | **Are the inclusion/exclusion criteria measured using valid and reliable measures?—RTI2** [*PI: Separately specify each criterion that abstractors should consider based on its relevance to study bias. It is unlikely that all criteria will need to be evaluated in relation to this question. Provide direction to abstractors on valid and reliable measurement of each criterion that is to be considered. For example, prior exposure or disease status is a frequent inclusion/exclusion criterion, particularly in inception cohorts. Subjective measures based on self-report tend to have lower reliability and validity than objective measures such as clinical reports and lab findings. Replicate question to evaluate each individual inclusion/exclusion criterion.* <br> *Note, in most cases firefighter studies will be coded yes since most studies rely upon administrative/employment/ death certificate records to identify firefighters for inclusion in the study.*] <br> Yes / No / Cannot determine; measurement approach not reported |
| | ***Did the study apply inclusion/exclusion criteria uniformly to all comparison groups/arms of the study?—RTI3*** [*PI: Drop question if not relevant to entire body of evidence (e.g., all case-series, single-arm studies).* <br> *Note: it may be possible that criteria are not uniformly applied when assembling firefighter cohorts or comparison groups*] <br> Yes / Partially: some, but not all criteria, applied to all arms or not clearly stated if some criteria are applied to all arms / No / Cannot determine: article does not specify / Not applicable: study has only one arm and so does not include comparison groups |
| | **Was the strategy for recruiting participants into the study the same across study groups/arms of the study?—RTI4** [*PIs: This question is likely to be more relevant for prospective or mixed designs than retrospective designs. Drop question if not relevant to entire body of evidence (e.g., all studies generally have only one arm).* <br> *Note: true recruitment into a cohort study is rare for firefighter cancer studies so most studies can be judged as meeting this criteria. In most cases for case-control studies this will be coded as "yes", unless case and control recruitment differs.*] <br> Yes / No / Cannot determine / Not applicable: one study group or arm |
| **INTERVENTIONS/EXPOSURE** | ***What is the level of detail in describing the intervention or exposure?—RTI5*** [*PI: Specify which details need to be stated (e.g., intensity, duration, frequency, route, setting, and timing of intervention/exposure). For case-control studies, consider whether the condition, timing, frequency, and setting of symptoms are provided in the case definition. PI needs to establish criteria for high, medium, or low response.* <br> *Note: Many firefighter exposures are often based on occupational title/certification/hospital-based record or death certificate which can be judged as medium given that it is a crude measure of exposure. Important to consider if the comparison group is truly unexposed or reasonably complete in order to mark as high*] <br> High: very clear, all PI-required details provided / Medium: somewhat clear, majority of PI-required details provided / Low: unclear, many PI-required details missing |

(*Continued*)

**Table 2.** (Continued)

| SECTION | STATEMENT, INSTRUCTIONS & SCORING |
|---|---|
| CREATION OF TREATMENT GROUPS | **Is the selection of the comparison group appropriate, after taking into account feasibility and ethical considerations.—RTI6** [*PI: Provide instruction to the abstractor based on the type of study. Interventions with community components are likely to have contamination if all groups are drawn from the same community. Interventions without community components should select groups from the same source (e.g., community or hospital) to reduce baseline differences across groups. For case-control studies, controls should represent the population from which cases arose; that is, controls should have met the case definition if they had the outcome.* <br> *Note: For most firefighter cohort studies the comparison group will be non-fighters drawn from the same registry while case control studies will draw controls from the same hospital system(s) or surrounding communities. In most cases these can be rated as "yes". Cases in which selected controls clearly do not reflect the same community in cohort studies can be rated as "no" (e.g., comparing deaths in firefighters in a particular state to nation-wide mortality). In cases where the time periods or exposure and disease are different across groups in comparison, rate as "no". In cases when multiple comparators/controls are applied select and rate the most favorable case (i.e., if at least one assessment is "yes" then report "yes for this item").*] <br> Yes / No / Cannot determine or no description of the derivation of the comparison group / Not applicable: study does not include a comparison group (case series, one study arm) |
| | **Any attempt to balance the allocation between the groups (e.g., through stratification, matching, propensity scores).—RTI7** [*PI: This is most likely to be used in case-control study designs. Drop if not relevant to the body of evidence.* <br> *Note: Score all prospective studies as "Not applicable"; for case-control studies that employ any of these in the design phase—not the analytic phase*] <br> Yes or study accounts for imbalance between groups through a post hoc approach such as multivariate analysis / No or cannot determine / Not applicable: study does not include a comparison group (case series or one study arm) |
| SOUNDNESS OF INFORMATION | **Are interventions/exposures assessed using valid and reliable measures, implemented consistently across all study participants?—RTI8** [*PI: Important measures may be listed separately. PI may need to establish a threshold for what would constitute acceptable measures based on study topic. When subjective or objective measures could be collected, subjective measures based on self- report may be considered as being less reliable and valid than objective measures such as clinical reports and lab findings. Replicate question when needed.* <br> *Note: Code High if there is a job title certification. If death certificate or registry records code medium, if no specification code Low.*] <br> High / Medium / Low |
| | **Are outcomes assessed using valid and reliable measures, implemented consistently across all study participants?—RTI9** <br> [*PI: Primary outcomes should be identified for abstractors and if there is more than one, they may be listed separately. Also, identify any relevant secondary outcomes and harms. Subjective measures based on self-report tend to have lower reliability and validity than objective measures such as clinical reports and lab findings. Note for case-control studies: consider whether the ascertainment of cases was independent of exposure.* <br> *Note: In most cases this will be coded "yes" since outcomes are typically derived registry/death certificate records and are collected in the same manner across exposure groups. In rare cases this information may not be collected in the same manner across exposure groups in which a "no" rating would be appropriate. Additional rare events could include poorly specified sources of incidence or mortality events which would warrant "Cannot determine. . .*] <br> Yes / No / Cannot determine or measurement approach not reported |
| FOLLOW-UP | **Is the length of follow-up the same for all groups?—RTI10** [*For case-control studies, are cases and controls matched on length of followup? Abstractor: When follow-up was the same for all study participants, the answer is yes. If different lengths of follow-up were adjusted by statistical techniques, (e.g., survival analysis), the answer is yes. Studies in which differences in follow-up were ignored should be answered no.* <br> *Note: Select "no" if exposures-outcome time-date imbalances vary across groups. Should be a lag of 2 years or more from exposure to outcome. If so, code Yes.*] <br> Yes / No or cannot determine / Not applicable: cross-sectional or only one group followed over time |
| | **Is the length of time following the intervention/exposure sufficient to support the evaluation of primary outcomes and harms?—RTI11** [*PI: Primary outcomes (including harms) should be identified for abstractors. Important measures may be listed separately. Abstractors should be provided with specific criteria for sufficient length of follow-up based on prior research or theory. Drop if entire body of evidence is cross-sectional or if minimal length of follow-up period is specified through inclusion criteria.* <br> *Note: If cross-sectional list non-applicable. Otherwise in most cases this will be coded "yes" if the follow-up period only includes events taking place at least two years after joining the cohort. If the follow-up period includes events taking place less than two years or after joining the cohort then code "No". If there is no information provided on the follow-up period then report "Cannot determine"*] <br> Yes / Partially: some primary outcomes are followed for a sufficient length of time / No / Cannot determine / Not applicable: cross-sectional |

(*Continued*)

**Table 2.** (Continued)

| SECTION | STATEMENT, INSTRUCTIONS & SCORING |
|---|---|
| ANALYSIS COMPARABILITY | **Were the important confounding and effect modifying variables taken into account in the design and/or analysis (e.g., through matching, stratification, interaction terms, multivariate analysis, or other statistical adjustment)?—RTI12** *[PI: Provide instruction to abstractors on adequate adjustment for confounding and testing for effect modification.* *Note: if only age is accounted for in the analysis or via age group stratification, then select "partially"; select "yes if two or more variables is accounted for in the analysis or via age group stratification; otherwise select "no"]* Yes / Partially: some variables taken into account or adjustment achieved to some extent / No: not accounted for or not identified / Cannot determine |
| INTERPRETATION | **Are results believable taking study limitations into consideration?—RTI13** *[Abstractor:This question is intended to capture the overall quality of the study. Consider issues that may limit your ability to interpret the results of the study. Review responses to earlier questions for specific criteria.* *Note: Most firefighter linkage studies have inherent limitations such as job title as proxy for exposures that limit study 'believability'. These studies and those employing case-control designs can also be impacted by limited statistical power. Coding of "partially" and "no" should be reserved for studies that have limitations beyond those that broadly impact firefighter cancer studies.]* Yes / Partially / No |
| PRESENTATION AND REPORTING | **Is the source of funding identified?** *[PI: The relevance of this question will depend upon the topic. This question may be modified to identify particular sources of funding (e.g., industry, government, university, or foundation funding).]* Yes / No |

***Note.*** Any modification to the original scale is italicized.

## Model specification

The FACETS computer program [48, 49] for Rasch analysis, was used to examine the quality of two study quality assessments using a Many-facet Rating Scale Model (MFRM) [48]. The MFRM is expressed as below.

$$ln\left[\frac{P_{jnmi,k}}{P_{jnmi,(k-1)}}\right] = \theta_j - \delta_i - \tau_k \qquad (3)$$

, where

$P_{jnmi,k}$ = probability of study $j$ receiving a rating $k$ on item $i$;
$P_{jnmi,(k-1)}$ = probability of study $j$ receiving a rating $k$-$1$ on item $i$;
$\theta_j$ = quality measure of study $j$;
$\delta_i$ = difficulty of endorsing item $i$;
$\tau_k$ = difficulty of endorsing category $k$ relative to $k$-$1$;

## Analyses

The ratio between $P_{jnmi,k}$ and $P_{jnmi,(k-1)}$ specified in Eq 3 is called odds so that the log-odds (logits) are a linear combination of latent measures for different facets. Since all the measures are on a common scale with logits as the units, the MFRM can create measures on an additive interval scale. Higher logit values reflect higher quality for studies, and items that are more difficult to endorse. These values were presented using a Wright map to show an empirical display of study quality scores and item difficulties.

In addition to logit values, we computed the reliability of separation indices for items; and study, rater, and Infit and Outfit MnSq statistics. The reliability of separation indices shows how reproducible the scale would be if using a different but equivalent study sample. Infit and Outfit MnSq statistics are used to demonstrate how well item and study fit the latent scale. Lastly, a Chi-square test is performed to examine if all items can be viewed as equal. A significant result indicates that the studies are distinct from each other.

## Results

### Descriptive statistics of study quality scale scores for firefighters' cancer literature

Table 3 displays the summary statistics of observed and Rasch scores for each of two study quality measures: (1) RTI and (2) NOQAS. The RTI scale items had an observed mean of 1.51 ($SD$ = .66) and a Rasch score mean of 0.00 ($SD$ = 3.85), while the items in NOQAS had an observed mean of 1.84 ($SD$ = .37) and a Rasch score mean of 0.00 ($SD$ = 2.29). In addition, the overall quality scores of 49 studies had an observed mean of 1.48 ($SD$ = 0.24) and a Rasch mean of 1.32 ($SD$ = 2.24) when measured by RTI scale, but they had an observed mean of 1.84 ($SD$ = 0.17) and a Rasch mean of 0.21 ($SD$ = 0.78), when measured by the NOQAS. These results indicate that on average, the RTI scale item banks produced much higher quality scores across 49 studies, when compared to the NOQAS.

Figs 1, 2 display the wright map, which is an empirical display of the RTI scale (Fig 1) and NOQAS (Fig 2), respectively.

**Table 3. Summary statistics.**

| Facets | | RTI | NOQAS |
|---|---|---|---|
| *Items* | | | |
| Observed Scores | $M$ | 1.51 | 1.84 |
| | $SD$ | 0.66 | 0.37 |
| Rasch Measures | $M$ | 0.00 | 0.00 |
| | $SD$ | 3.85 | 2.29 |
| Infit MnSq | $M$ | 1.13 | 1.08 |
| | $SD$ | 0.52 | 0.33 |
| Outfit MnSq | $M$ | 0.84 | 0.93 |
| | $SD$ | 0.40 | 0.40 |
| *Studies* | | | |
| Observed Scores | $M$ | 1.48 | 1.84 |
| | $SD$ | 0.24 | 0.17 |
| Rasch Measures | $M$ | 1.32 | 0.21 |
| | $SD$ | 2.24 | 0.78 |
| Infit MnSq | $M$ | 1.00 | 0.90 |
| | $SD$ | 1.00 | 0.50 |
| Outfit MnSq | $M$ | 0.85 | 0.92 |
| | $SD$ | 1.14 | 0.95 |
| *Raters* | | | |
| Observed Scores | $M$ | 1.51 | 1.84 |
| | $SD$ | 0.66 | 0.05 |
| Rasch Measures | $M$ | .00 | .00 |
| | $SD$ | 3.85 | 0.20 |
| Infit MnSq | $M$ | 1.13 | 0.99 |
| | $SD$ | 0.52 | 0.24 |
| Outfit MnSq | $M$ | 0.85 | 0.94 |
| | $SD$ | 0.40 | 0.46 |

*Note*. RTI: Item Bank on Risk of Bias and Precision of Observational Studies developed by the RTI International; NOQAS: Newcastle-Ottawa Quality Assessment Scale; MnSq: Mean Square Error

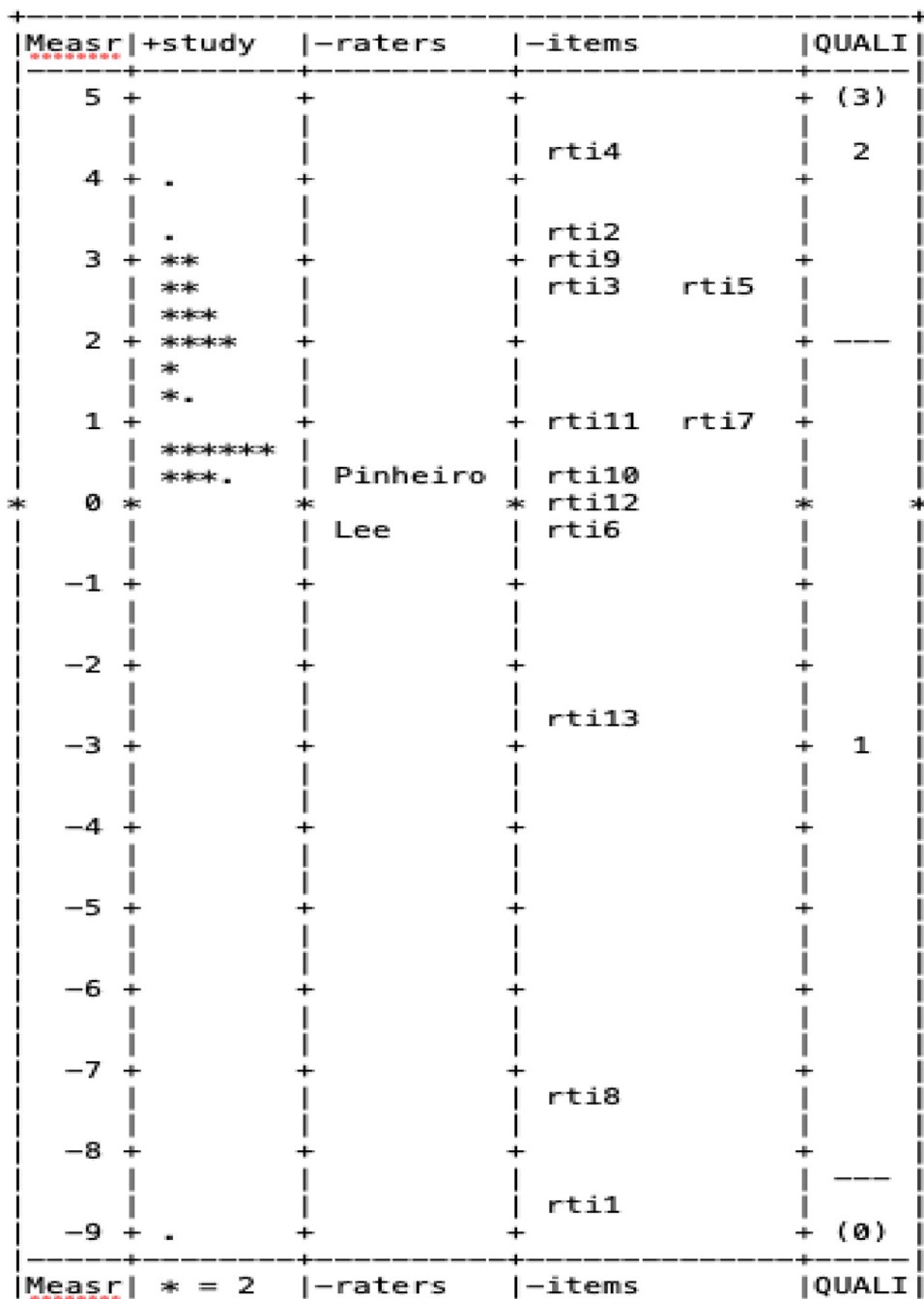

**Fig 1. Wright map for risk of bias and precision of observational studies.**

In each figure, the first column shows the Rasch score on a logit scale. The items (column 4), raters (column 3), and the individual studies (column 2) are located on the wright map based on their Rasch score. The last column displays threshold estimates of response categories on the Likert scale. As shown in Fig 1, the latent Rasch scores of study quality measured by the

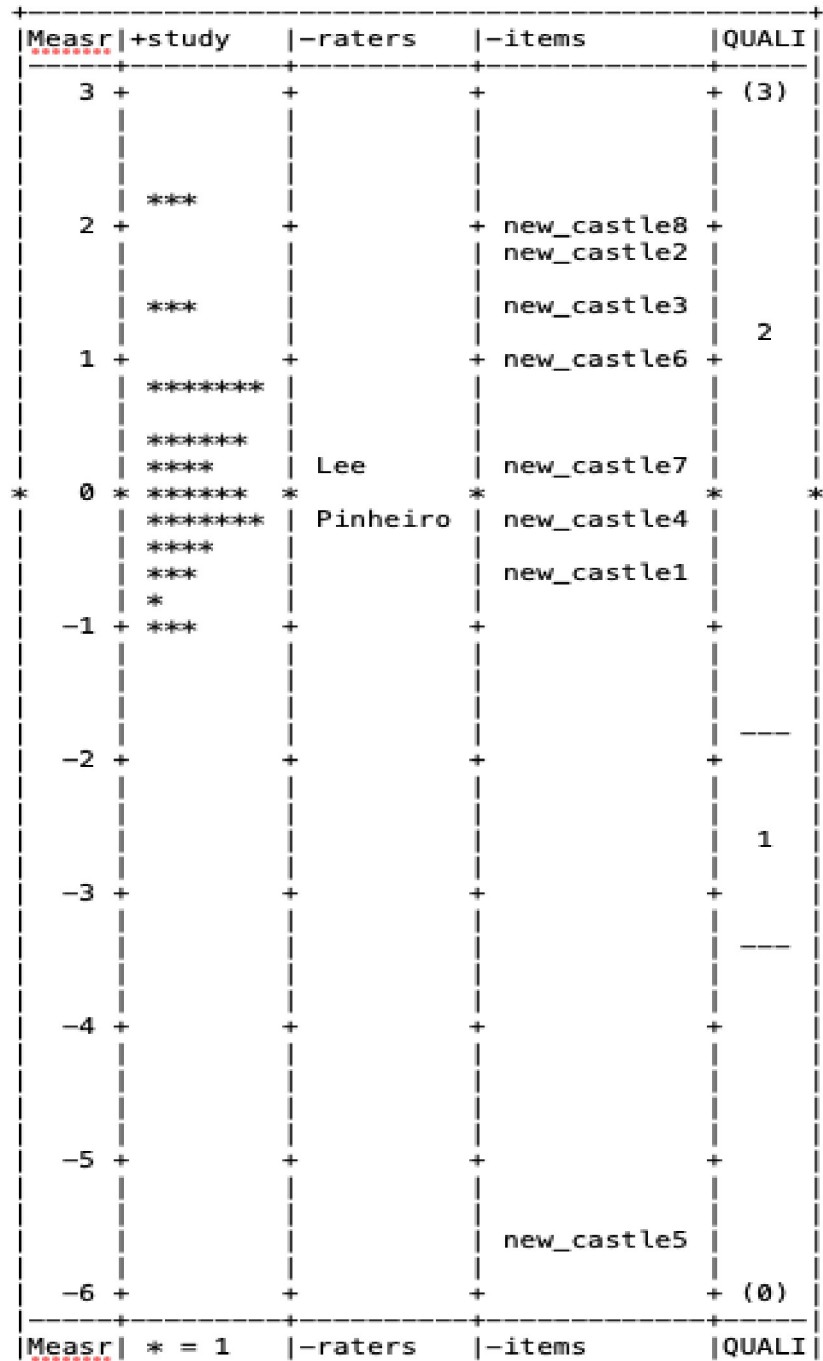

**Fig 2. Wright map for Newcastle-Ottawa quality assessment scale.**

RTI scale were skewed to the left, indicating that most studies appeared to be low in its study quality, while those Rasch scores measured by NOQAS followed a normal distribution (see Fig 2). Out of 13 items in the RTI scale item bank, 9 were above the mean of 0 (column 3 in Fig 1), indicating that most were quite difficult to evaluate. On the other hand, items on NOQAS

were distributed relatively evenly in terms of item difficulty (column 3 in Fig 2), except item #5 (very easy; located at 5 standard deviations below the mean). Lastly, two raters (column 2) were quite consistent in evaluating the quality of primary studies using the NOQAS and RTI scale item bank in their ratings of study quality.

### Psychometric evidence for the RTI items for firefighters' cancer literature

**Dimensionality.** Results from the Many-facet Rating Scale Model (MFRM) indicated that there was one underlying factor that explained 79.82% of variances in 13 items. This result suggests that the RTI is unidimensional for measuring the quality of individual studies (> 20%) [49].

**Reliability.** The reliability of separation for RTI scale items was .99 (near 1.0), implying that the distribution of item measures can well represent the entire range of latent scale. The reliability value higher than .80 suggest that the RTI scale item bank scale does present acceptable reproducibility and consistency of the ordering of the Rasch scale scores.

**Validity.** As shown in Table 4, the Infit and Outfit MnSq for RTI scale items 6, 9, 10, and 12 were found to fall into a fit category of A, indicating a good fit of each item to the study quality scale. Although items 2 and 11 fell into a fit category of A using the Infit MnSq, item 11 did show high MnSq. Items 1, 3, 5, 7, 8 and 11 were less productive based on either Outfit or Infit MnSq. See Table 4.

The quality measures of the 49 studies were significantly different, $\chi^2$ (46) = 177.7, $p < .01$. As shown in Table 5, study 29 had the highest study quality score, while study 8 had the lowest. Most studies fit the scale well with a fit category of A or B, with six studies falling into category C or D.

### Psychometric evidence for the NOQAS items for firefighters' cancer literature

**Dimensionality.** The result from MFRM indicated that one underlying factor exists. This factor explained 41.41% of variances in 8 items, suggesting that the NOQAS is unidimensional for measuring the quality of individual studies (> 20%) [49].

**Table 4. Summary of Rasch measures for RTI risk of bias and precision of observational studies.**

| Item Number | Observed Score | Rasch Measures | SE | Infit | | Outfit | |
|---|---|---|---|---|---|---|---|
| | | | | MnSq | Category | MnSq | Category |
| 1 | 2.97 | -8.52 | 0.59 | 1.70 | C | 0.94 | A |
| 8 | 2.92 | -7.45 | 0.38 | 1.90 | C | 1.64 | C |
| 13 | 1.94 | -2.53 | 0.21 | 0.40 | B | 0.40 | B |
| 6 | 1.48 | -.38 | 0.22 | 1.15 | A | 1.14 | A |
| 12 | 1.38 | 0.11 | 0.23 | 0.92 | A | 0.92 | A |
| 10 | 1.39 | 0.24 | 0.23 | 1.06 | A | 0.91 | A |
| 11 | 1.23 | 0.92 | 0.26 | 0.21 | A | 0.97 | C |
| 7 | 1.20 | 1.13 | 0.27 | 1.06 | B | 0.99 | D |
| 3 | 1.05 | 2.82 | 0.48 | 1.70 | C | 0.66 | A |
| 5 | 1.05 | 2.82 | 0.48 | 1.77 | C | 0.87 | A |
| 9 | 1.04 | 3.06 | 0.53 | 1.01 | A | 1.16 | A |
| 2 | 1.03 | 3.35 | 0.59 | 0.86 | A | 0.37 | B |
| 4 | 1.00 | 4.43 | 0.85 | 0.01 | B | 0.01 | B |

Note. Observed average indicates mean score of all response ratings for each item. Rasch measure reflects the location of an item on the Rasch scale. S.E. stands for standard error for each Rasch measure. The Infit and Outfit mean square error (MSE) are fit indices for identifying misfit items. Mean Square Error (MnSq).

**Table 5.  Summary for study quality scores by RTI risk of bias and precision of observational studies.**

| Study | Observed | Rasch | SE | Infit | | Outfit | |
|---|---|---|---|---|---|---|---|
| ID | Score | Measures | | MnSq | Category | MnSq | Category |
| 8 | 0 | -12.77 | 1.99 | - | - | - | - |
| 13 | 1.38 | 0.33 | 0.71 | 0.21 | B | 0.1 | B |
| 14 | 1.38 | 0.33 | 0.71 | 0.21 | B | 0.1 | B |
| 16 | 1.38 | 0.33 | 0.71 | 0.21 | B | 0.1 | B |
| 25 | 1.38 | 0.33 | 0.71 | 0.21 | B | 0.1 | B |
| 37 | 1.38 | 0.33 | 0.71 | 0.21 | B | 0.1 | B |
| 38 | 1.38 | 0.33 | 0.71 | 5.1 | D | 4.29 | D |
| 40 | 1.38 | 0.33 | 0.71 | 0.21 | B | 0.1 | B |
| 1 | 1.42 | 0.81 | 0.67 | 0.37 | B | 0.17 | B |
| 2 | 1.42 | 0.81 | 0.67 | 0.37 | B | 0.17 | B |
| 5 | 1.42 | 0.81 | 0.67 | 2.8 | D | 2.28 | D |
| 7 | 1.42 | 0.81 | 0.67 | 0.5 | A | 0.3 | B |
| 11 | 1.42 | 0.81 | 0.67 | 0.56 | A | 0.5 | A |
| 21 | 1.42 | 0.81 | 0.67 | 0.37 | B | 0.17 | B |
| 22 | 1.42 | 0.81 | 0.67 | 0.37 | B | 0.17 | B |
| 23 | 1.42 | 0.81 | 0.67 | 0.37 | B | 0.17 | B |
| 28 | 1.42 | 0.81 | 0.67 | 0.57 | A | 0.55 | A |
| 43 | 1.42 | 0.81 | 0.67 | 3.41 | D | 5.36 | D |
| 48 | 1.42 | 0.81 | 0.67 | 0.47 | B | 0.26 | B |
| 49 | 1.42 | 0.81 | 0.67 | 0.47 | B | 0.26 | B |
| 12 | 1.46 | 1.23 | 0.63 | 0.54 | A | 0.28 | B |
| 19 | 1.46 | 1.23 | 0.63 | 0.56 | A | 0.3 | B |
| 45 | 1.46 | 1.23 | 0.63 | 0.51 | A | 0.25 | B |
| 6 | 1.5 | 1.62 | 0.6 | 0.6 | A | 0.3 | B |
| 47 | 1.5 | 1.62 | 0.6 | 0.8 | A | 1.42 | A |
| 3 | 1.54 | 1.96 | 0.57 | 0.6 | A | 0.31 | B |
| 18 | 1.54 | 1.96 | 0.57 | 0.64 | A | 0.33 | B |
| 26 | 1.54 | 1.96 | 0.57 | 1.05 | A | 1.35 | A |
| 31 | 1.54 | 1.96 | 0.57 | 0.62 | A | 0.32 | B |
| 32 | 1.54 | 1.96 | 0.57 | 0.75 | A | 0.45 | B |
| 33 | 1.54 | 1.96 | 0.57 | 0.97 | A | 0.58 | A |
| 46 | 1.54 | 1.96 | 0.57 | 0.65 | A | 0.37 | B |
| 34 | 1.56 | 2.11 | 0.58 | 0.92 | A | 0.51 | A |
| 10 | 1.58 | 2.27 | 0.55 | 1.21 | A | 0.64 | A |
| 20 | 1.58 | 2.27 | 0.55 | 0.86 | A | 0.49 | B |
| 27 | 1.58 | 2.27 | 0.55 | 1.18 | A | 1.5 | A |
| 35 | 1.58 | 2.27 | 0.55 | 0.56 | A | 0.31 | B |
| 36 | 1.58 | 2.27 | 0.55 | 0.52 | A | 0.29 | B |
| 44 | 1.58 | 2.27 | 0.55 | 3.03 | D | 2.97 | D |
| 17 | 1.62 | 2.56 | 0.53 | 0.6 | A | 0.35 | B |
| 41 | 1.62 | 2.56 | 0.53 | 1.08 | A | 0.59 | A |
| 42 | 1.62 | 2.56 | 0.53 | 1.33 | A | 0.74 | A |
| 4 | 1.64 | 2.57 | 0.53 | 2.85 | D | 3.84 | D |
| 9 | 1.65 | 2.83 | 0.52 | 0.75 | A | 0.45 | B |
| 15 | 1.65 | 2.83 | 0.52 | 0.93 | A | 1.08 | A |
| 39 | 1.65 | 2.83 | 0.52 | 1.84 | C | 1.43 | A |

*(Continued)*

**Table 5.** (Continued)

| Study | Observed | Rasch | SE | Infit | | Outfit | |
|---|---|---|---|---|---|---|---|
| ID | Score | Measures | | MnSq | Category | MnSq | Category |
| 24 | 1.69 | 3.1 | 0.51 | 1.29 | A | 1.09 | A |
| 30 | 1.73 | 3.35 | 0.5 | 0.55 | A | 0.42 | B |
| 29 | 1.88 | 4.09 | 0.48 | 3.1 | D | 2.47 | D |

Note. Observed average indicates mean score of all response ratings for each study. Rasch measure reflects the location of an item on the Rasch scale. S.E. stands for standard error for each Rasch measure. The Infit and Outfit mean square error (MSE) are fit indices for identifying misfit studies; A list of studies can be found in S1 Appendix. Mean Square Error (MnSq).

**Reliability.**   The reliability of separation for items was .99, implying that the NOQAS quality assessment scale does show acceptable reproducibility and consistency of the ordering of the Rasch scale scores.

**Validity.**   As shown in Table 6, the Infit and Outfit MnSq for most items fell into a fit category of A, indicating a good fit of each item to the study quality scale. Exceptions were item 1 (B for Outfit MnSq), item 7 (C for Infit and Outfit MnSq). Particularly, item 7 had high Infit and Outfit MnSq values, showing that this item is unproductive and distorting. Content specialists should be consulted in terms of future uses of item 7 for this scale.

Results indicated that the study quality measures of these 49 studies were significantly different, $\chi^2$ (46) = 71.9, $p$ = .05. As shown in Table 7, study 12 had the highest study quality score, while study 40 had the lowest. Most studies fit the scale well with a fit category of A or B, with eight studies showing falling into category C or D.

## Comparison between RTI and NOQAS for firefighters' cancer literature

The reliability of item separation index for the RTI scale and NOQAS were both found to be high (approximating 1), indicating that both scales are reproducible and consistent of the ordering of Rasch scale scores. In terms of rater agreement, the two coders rated study quality equally consistent using NOQAS than RTI, as shown in Tables 8 and 9. The NOQAS measures produced much lower Rasch latent study quality scores with a less variation, and their items

**Table 6. Summary of Rasch measures for Newcastle-Ottawa quality assessment scale.**

| Item Number | Observed Score | Rasch Measures | SE | Infit | | Outfit | |
|---|---|---|---|---|---|---|---|
| | | | | MnSq | Category | MnSq | Category |
| 5 | 2.66 | -5.56 | 0.23 | 1.47 | A | 1.30 | A |
| 1 | 1.98 | -0.58 | 0.41 | 0.65 | A | 0.33 | B |
| 4 | 1.96 | -0.29 | 0.38 | 1.11 | A | 0.93 | A |
| 7 | 1.91 | 0.16 | 0.31 | 1.52 | C | 1.73 | C |
| 3 | 1.76 | 1.02 | 0.21 | 1.18 | A | 0.68 | A |
| 6 | 1.67 | 1.32 | 0.18 | 0.96 | A | 0.82 | A |
| 2 | 1.45 | 1.86 | 0.15 | 1.18 | A | 1.00 | A |
| 8 | 1.35 | 2.07 | 0.15 | 0.55 | A | 0.67 | A |

*Note*. Observed average indicates mean score of all response ratings for each item. Rasch measure reflects the location of an item on the Rasch scale. S.E. stands for standard error for each Rasch measure. The Infit and Outfit mean square error (MSE) are fit indices for identifying misfit items. Mean Square Error (MnSq).

**Table 7. Summary for study quality scores by NOQAS scale.**

| Study ID | Observed Score | Rasch Measures | SE | Infit | | Outfit | |
|---|---|---|---|---|---|---|---|
| | | | | MnSq | Category | MnSq | Category |
| 40 | 1.44 | -1.07 | 0.38 | 1.06 | A | 0.8 | A |
| 24 | 1.50 | -0.92 | 0.39 | 1.09 | A | 0.89 | A |
| 30 | 1.50 | -0.92 | 0.39 | 1.19 | A | 0.96 | A |
| 35 | 1.56 | -0.77 | 0.39 | 2.16 | D | 3.76 | D |
| 17 | 1.63 | -0.61 | 0.4 | 0.65 | A | 0.49 | B |
| 43 | 1.63 | -0.61 | 0.4 | 0.66 | A | 0.56 | A |
| 48 | 1.63 | -0.61 | 0.4 | 0.9 | A | 0.66 | A |
| 15 | 1.69 | -0.45 | 0.42 | 0.8 | A | 0.6 | A |
| 34 | 1.69 | -0.45 | 0.42 | 1.32 | A | 0.92 | A |
| 42 | 1.69 | -0.45 | 0.42 | 1.77 | C | 2 | C |
| 47 | 1.69 | -0.45 | 0.42 | 0.33 | B | 0.32 | B |
| 4 | 1.75 | -0.26 | 0.44 | 2.02 | C | 2.15 | D |
| 9 | 1.75 | -0.26 | 0.44 | 1.17 | A | 0.74 | A |
| 18 | 1.75 | -0.26 | 0.44 | 1.15 | A | 2.62 | D |
| 27 | 1.75 | -0.26 | 0.44 | 0.84 | A | 0.55 | A |
| 28 | 1.75 | -0.26 | 0.44 | 0.84 | A | 0.55 | A |
| 31 | 1.75 | -0.26 | 0.44 | 0.51 | A | 0.43 | B |
| 33 | 1.75 | -0.26 | 0.44 | 1.43 | A | 1.02 | A |
| 20 | 1.75 | -0.07 | 0.61 | 0.8 | A | 0.53 | A |
| 10 | 1.81 | -0.06 | 0.46 | 1.55 | C | 3.35 | D |
| 36 | 1.81 | -0.06 | 0.46 | 0.72 | A | 0.49 | B |
| 37 | 1.81 | -0.06 | 0.46 | 1.02 | A | 0.87 | A |
| 19 | 1.88 | -0.05 | 0.71 | 0.47 | B | 0.39 | B |
| 25 | 1.88 | -0.05 | 0.71 | 0.47 | B | 0.39 | B |
| 3 | 1.88 | 0.17 | 0.5 | 1.53 | C | 2.63 | D |
| 5 | 1.88 | 0.17 | 0.5 | 1.47 | A | 0.83 | A |
| 23 | 1.88 | 0.17 | 0.5 | 1.01 | A | 0.7 | A |
| 39 | 1.88 | 0.17 | 0.5 | 1.26 | A | 0.9 | A |
| 1 | 1.94 | 0.45 | 0.56 | 1.65 | C | 3.28 | D |
| 2 | 1.94 | 0.45 | 0.56 | 1.65 | C | 3.28 | D |
| 22 | 1.94 | 0.45 | 0.56 | 0.46 | B | 0.28 | B |
| 41 | 1.94 | 0.45 | 0.56 | 0.42 | B | 0.24 | B |
| 46 | 1.94 | 0.45 | 0.56 | 0.6 | A | 0.48 | B |
| 50 | 1.94 | 0.45 | 0.56 | 0.48 | B | 0.33 | B |
| 11 | 2.0 | 0.81 | 0.66 | 1.41 | A | 0.66 | A |
| 16 | 2.0 | 0.81 | 0.66 | 0.54 | A | 0.29 | B |
| 21 | 2 | 0.81 | 0.66 | 0.49 | B | 0.24 | B |
| 26 | 2 | 0.81 | 0.66 | 0.54 | A | 0.29 | B |
| 32 | 2 | 0.81 | 0.66 | 0.7 | A | 0.51 | A |
| 38 | 2 | 0.81 | 0.66 | 0.54 | A | 0.29 | B |
| 49 | 2 | 0.81 | 0.66 | 0.66 | A | 0.59 | A |
| 13 | 2.06 | 1.34 | 0.81 | 0.6 | A | 0.38 | B |
| 14 | 2.06 | 1.34 | 0.81 | 0.6 | A | 0.38 | B |
| 29 | 2.06 | 1.34 | 0.81 | 0.6 | A | 0.38 | B |
| 6 | 2.13 | 2.12 | 0.99 | 0.05 | B | 0.04 | B |
| 7 | 2.13 | 2.12 | 0.99 | 0.05 | B | 0.04 | B |

*(Continued)*

**Table 7.** (Continued)

| Study ID | Observed Score | Rasch Measures | SE | Infit | | Outfit | |
|---|---|---|---|---|---|---|---|
| | | | | MnSq | Category | MnSq | Category |
| 12 | 2.13 | 2.12 | 0.99 | 0.05 | B | 0.04 | B |

*Note.* Observed average indicates mean score of all response ratings for each study. Rasch measure reflects the location of a study on the Rasch scale. The Infit and Outfit mean square error (MSE) are fit indices for identifying misfit items. Mean Square Residual (MnSq). A list of studies can be found in S1 Appendix.

**Table 8. Summary for rater score by RTI risk of bias and precision of observational studies.**

| Rater | Observed | Rasch | SE | Infit | | Outfit | |
|---|---|---|---|---|---|---|---|
| Number | Score | Measures | | MnSq | Category | MnSq | Category |
| 1 | 1.53 | -0.17 | 0.12 | 1.16 | A | 1.07 | A |
| 2 | 1.49 | 0.17 | 0.12 | 0.92 | A | 0.62 | A |

*Note.* Observed average indicates mean score of all response ratings by each rater. Rasch measure reflects the location of a rater on the Rasch scale. The Infit and Outfit mean square error (MSE) are fit indices. Mean Square Error (MnSq).

**Table 9. Summary for rater scores by NOQAS scale.**

| Rater ID | Observed Score | Rasch Measures | SE | Infit | | Outfit | |
|---|---|---|---|---|---|---|---|
| | | | | MnSq | Category | MnSq | Category |
| Rater 1 | 1.79 | 0.20 | 0.10 | 1.23 | A | 1.40 | A |
| Rater 2 | 1.89 | -0.20 | 0.11 | 0.75 | A | 0.48 | B |

*Note.* Observed average indicates mean score of all response ratings by each rater. Rasch measure reflects the location of a rater on the Rasch scale. The Infit and Outfit mean square error (MSE) are fit indices. Mean Square Error (MnSq)

were much easier to rate than the RTI. The NOQAS had more items that showed better fit between items and the overall quality scores. The reason for this could be that the NOQAS was adapted to assess the quality of firefighters' cancer literature, which may have enabled ratings to be more closely aligned and less varied. As a result, there may be better fits between items and the quality scores. Additionally, study quality scores measured by the NOQAS scale were found to follow normal distribution. Both measures were found to be unidimensional.

## Discussion

Using firefighters' cancer literature, the current study is the first attempt to examine the psychometric properties of two commonly used study quality assessment measures using the Rasch measurement theory. Of many strengths, Rasch models can be used to (a) produce invariant study quality measures on a latent continuum, (b) assess the validity, reliability, and fairness of latent measures, and (c) use latent scores to explain variation in outcome measures. These characteristics of Rasch measurement theory offer practical applications in meta-analysis. Of many, study quality scores estimated by Rasch measurement model enable us to be directly compared across different studies and further modeled to explain variation in study effects by study quality scores.

Our study found that the RTI scale and NOQAS were reproducible and consistent in evaluating the quality of firefighters' cancer literature, showing higher item reliability. In terms of interrater reliability, two raters were quite consistent in their assessment of study quality, when using both RTI and NOQAS scales. In terms of validity, we found that the NOQAS has more items that show better fit to the underlying construct of study quality than the RTI scale. This result indicates that NOQAS demonstrates better validity of internal structure to measure the quality of firefighters' cancer literature. Lastly, latent scores measured using NOQAS were distributed across all range of the latent scores, with much lower study quality scores with a smaller variation. These results suggest that NOQAS items are much easier to rate the quality of firefighters' cancer literature. Our findings accord with a previous study conducted by Margulis and her colleague [40], which concludes that RTI was harder to apply and thus produces much heterogenous quality scores than NOQAS.

The present study is significant in at least two major respects. First, the current study is the first in its kind that assesses the psychometric properties—reliability and validity—for two quality assessment tools that are most used in observational studies. Previous studies focused on interrater reliability of NOQAS and RTI scales, thus leaving the item reliability and validity of NOQAS and RTI unanswered. The current study provides the psychometric properties—reliability and validity—of NOQAS and RTI for future use beyond interrater reliability. Second, more importantly, we used the Rasch Measurement theory (RMT) that produces the compatible quality scores of the included studies in meta-analysis, which further enhance its generalizability and applicability in meta-analysis. It is because that Rasch scores allow us to utilize parametric statistical analysis, which mostly assumes normal distribution. When utilizing the Rasch scores of NOQAS and RTI in a meta-analysis of firefighters' cancer incidence and mortality, we found that NOQAS scores significantly predict variation in the effect sizes. Specifically, results from a mixed-effects model indicate a significant and positive relationship between quality score and firefighters' cancer incidence and mortality. Lastly, the item parameters estimated by RMT are generally invariant to the population, which will offer greater generalization of meta-analytic results.

In this study, we did not address one of the important psychometric properties: whether NOQAS and RTI showed fairness in its assessment. If NOQAS and RTI are equally applicable to any study, it is expected that NOQAS and RTI scores are invariant regardless of study characteristics such as sampling method, funding sources, inclusiveness of samples, and whether a study used a good-quality instrument or not. Despite this limitation, the current study certainly adds to our understanding of the psychometric properties of NOQAS and RTI for future meta-analyses of the observational studies, similar to firefighters' cancer literature.

## Supporting information

**S1 Appendix. Included 49 studies.**
(DOCX)

## Author Contributions

**Conceptualization:** Soyeon Ahn, David J. Lee.

**Data curation:** Soyeon Ahn, Paulo S. Pinheiro, Laura A. McClure, David J. Lee.

**Formal analysis:** Soyeon Ahn.

**Investigation:** Soyeon Ahn.

**Methodology:** Soyeon Ahn, Paulo S. Pinheiro.

**Project administration:** Soyeon Ahn.

**Resources:** Soyeon Ahn.

**Software:** Soyeon Ahn.

**Supervision:** Soyeon Ahn.

**Validation:** Soyeon Ahn.

**Visualization:** Soyeon Ahn.

**Writing – original draft:** Soyeon Ahn, David J. Lee.

**Writing – review & editing:** Soyeon Ahn, Paulo S. Pinheiro, Laura A. McClure, Diana R. Hernandez, Alberto J. Caban-Martinez, David J. Lee.

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
