## [Decision Letter · Decision Letter 0]

24 Feb 2023

PONE-D-22-30598An Examination of Psychometric Properties of Study Quality Assessment Scales in Meta-analysis: Rasch Measurement Model Applied to the Firefighter Cancer LiteraturePLOS ONE

Dear Dr. AHN,

Thank you for submitting your manuscript to PLOS ONE. After careful consideration, we feel that it has merit but does not fully meet PLOS ONE’s publication criteria as it currently stands. Therefore, we invite you to submit a revised version of the manuscript that addresses the points raised during the review process.

 expected to include results but may include pilot data. 

We look forward to receiving your revised manuscript.

Kind regards,

Simon Grima, PhD

Academic Editor

PLOS ONE

Journal Requirements:

Reviewers' comments:

Reviewer's Responses to Questions

**Comments to the Author**

1. Is the manuscript technically sound, and do the data support the conclusions?

Reviewer #1: Yes

Reviewer #2: Yes

2. Has the statistical analysis been performed appropriately and rigorously? 

Reviewer #1: I Don't Know

Reviewer #2: Yes

3. Have the authors made all data underlying the findings in their manuscript fully available?

Reviewer #1: Yes

Reviewer #2: Yes

4. Is the manuscript presented in an intelligible fashion and written in standard English?

Reviewer #1: Yes

Reviewer #2: Yes

5. Review Comments to the Author

Reviewer #1: This is a nicely done paper examining psychometric properties of two study quality assessment tools used in systematic reviews and meta-analysis. I am not familiar with Rasch measures or the RMT, so was not able to assess the statistical components of the analysis. However, I overall think this study is well conducted and would be a useful addition to the literature, as there are few such evaluations of the various quality assessment tools out there.

I have a few more substantial questions which I think, if addressed, might help strengthen the manuscript:

1. The NOQAS, by design, is meant to be adapted to different topics, so is functionally not the same scale for different users. The authors adapted it to their topic (firefighters/cancer), which was appropriate. However, I think this aspect should be discussed when comparing psychometric properties of this tool.

2. I think it would be helpful to add a sentence describing the training/characteristics of coders for this systematic review – this will help in understanding how easy/hard it was for them to apply these quality assessment tools.

3. The authors note that of 49 studies included in their review, only 47 were able to be evaluated by raters using NOQAS (page 7 in the second start of page numbering). Why?

I also noticed a few minor suggestions for correction as I read through the manuscript:

1. Can the Rasch measurement model be cited when first mentioned in the introduction? (Page 3)

2. PsycINFO does not have an H in it. (Page 6)

3. Need to add a (1) before standardized incidence ratio. (Page 10 if the page numbering were consistent)

4. Table 7 is a bit long and may be more helpful as supplemental material.

5. Discussion: Need to add citation to Margulis and her colleague(s) – page 11 in the new numbering.

6. Some reference formatting looks like it got off (perhaps with reference management software) so should be double-checked prior to publication.

Reviewer #2: Please add the more latest citation in the paper.

Compared the discussion with latest exiting literature.

In Detailed, Please explained the Research gap.

Please Add the research hypothesis and Research Question.

Check the style of the reference according to the requirement of the journal.

6. PLOS authors have the option to publish the peer review history of their article (what does this mean?). If published, this will include your full peer review and any attached files.

Reviewer #1: **Yes: **Caitlin E. Kennedy

Reviewer #2: **Yes: **sanjay taneja

While revising your submission, please upload your figure files to the Preflight Analysis and Conversion Engine (PACE) digital diagnostic tool, https://pacev2.apexcovantage.com/. PACE helps ensure that figures meet PLOS requirements. To use PACE, you must first register as a user. Registration is free. Then, login and navigate to the UPLOAD tab, where you will find detailed instructions on how to use the tool. If you encounter any issues or have any questions when using PACE, please email PLOS at figures@plos.org. Please note that Supporting Information files do not need this step.<quillbot-extension-portal></quillbot-extension-portal>

---

## [Author Response · Author response to Decision Letter 0]

30 Mar 2023

Dear Editor and Reviewers:

Thank you for the great feedbacks on our paper, which have improved our paper tremendously. Please find our responses on each comment made by editors and reviewers in a table added to the "response to reviewers" document. We are looking forward to publishing our paper in PLOS ONE. 

Thank you so much.

---

## [Editor Report · Decision Letter 1]

3 Apr 2023

An Examination of Psychometric Properties of Study Quality Assessment Scales in Meta-analysis: Rasch Measurement Model Applied to the Firefighter Cancer Literature

PONE-D-22-30598R1

Dear Dr. AHN,

We’re pleased to inform you that your manuscript has been judged scientifically suitable for publication and will be formally accepted for publication once it meets all outstanding technical requirements.

Kind regards,

Simon Grima, PhD

Academic Editor

PLOS ONE

Additional Editor Comments (optional):

Reviewers' comments:

<quillbot-extension-portal></quillbot-extension-portal>

---

## [Editor Report · Acceptance letter]

11 Apr 2023

PONE-D-22-30598R1 

An examination of psychometric properties of study quality assessment scales in meta-analysis: Rasch measurement model applied to the firefighter cancer literature 

Dear Dr. Ahn:

I'm pleased to inform you that your manuscript has been deemed suitable for publication in PLOS ONE. Congratulations! Your manuscript is now with our production department. 

Kind regards, 

on behalf of

Professor Simon Grima 

Academic Editor

PLOS ONE